# Individualised Halo-Free Gradient-Domain Colour Image Daltonisation

**DOI:** 10.3390/jimaging6110116

**Published:** 2020-10-29

**Authors:** Ivar Farup

**Affiliations:** Department of Computer Science, Norwegian University of Science and Technology (NTNU), 2815 Gjøvik, Norway; ivar.farup@ntnu.no

**Keywords:** daltonisation, colour vision deficiencies, anisotropic diffusion

## Abstract

Daltonisation refers to the recolouring of images such that details normally lost by colour vision deficient observers become visible. This comes at the cost of introducing artificial colours. In a previous work, we presented a gradient-domain colour image daltonisation method that outperformed previously known methods both in behavioural and psychometric experiments. In the present paper, we improve the method by (i) finding a good first estimate of the daltonised image, thus reducing the computational time significantly, and (ii) introducing local linear anisotropic diffusion, thus effectively removing the halo artefacts. The method uses a colour vision deficiency simulation algorithm as an ingredient, and can thus be applied for any colour vision deficiency, and can even be individualised if the exact individual colour vision is known.

## 1. Introduction

There are three types of cone cells, or cones, each with different pigment, in the retinas of the human eye [1]. The cones are refered to as L-cones, M-cones, and S-cones, suggesting sensitivities to long, medium, and short wavelenghts, respectively. Among colour normal observers (trichromats) there are only minor variations in the spectral sensitivities of the three cone types. However, about eight percent of the male population and 0.4 percent of the female population suffer from a congenital colour vision deficiency (CVD) [2]. This is manifested either as a change in one of the cone types (anomalous trichromacy), or as a complete lack of one (dichromacy) or, very seldomly, two (monochromacy) of the cone types. Referring to the affected L, M, and S-cones with the Greek prefixes prot-, deuter-, and tri-, respectively, these CVDs can be further characterised as protanomaly, deuteranomaly, and tritanomaly for the anomalous trichromacies, and protanopia, deuteranopia, and tritanopia for the dichromacies. Anomalies in or absence of the L or M cones both lead to various forms of red–green CVDs, whereas anomalies in of absence of the S cones lead to blue–yellow CVDs. In the population, the various forms of red–green CVDs are by far the most common ones.

In many situations, e.g., in graphic design, it can be important for colour normal observers to understand how the world looks through the eyes of a CVD observer. Various simulation methods have been developed for this purpose. In a thorough behavioural comparison of existing CVD simulation methods performed by Simon-Liedtke and Farup [3], the method of Brettel et al. [4] was found to be the best performing one for dichromats.

For the most part, people with CVDs can live completely normal lives. However, in some situations, they may experience certain disadvantages [5]. A considerable effort has been undertaken for coming up with techniques to improve the ability of CVD people to differentiate colour better through optical filters (https://enchroma.com/, visited 28 October 2020), introduction of lost information through sensory substitution devices [6], and even a medical ‘cure’ has been proposed [7]. For the display of digital images, so-called daltonisation techniques, i.e., recolouring of images for increased visibility for CVD observers, have been proposed [8,9,10,11]. CVD simulation algorithms are often an important ingredient in the daltonisation algorithms. Most daltonisation methods use some form of a global recolouring scheme, but daltonisation methods highlighting only colour edges and details that are lost to CVD observers have also been proposed [12]. In a behavioural experiment, the methods of Anagnostopoulos et al. [8] and Kotera [10] were found to be the best performing ones available at the time [13]. A thorough review of global daltonisation methods, i.e., mappings in colour space not taking the local image content into account, was recently given by Ribeiro and Gomes [14]. For image-dependent daltonisation, recent contributions include Chatzistamatis et al.’s method for recoloring of art paintings guided by semantic segmentation [15], and Iqbal et al.’s use of sharpened wavelet bands and inpainting [16].

Recently, we developed a daltonisation method that performed better or on par with state-of-the-art algorithms in both behavioural and psychometric experiments [17]. (An at least principally very similar method, although significantly differently formulated, was almost simultaneously proposed by Zhu et al. [18].) The method was based on a global transformation in the gradient domain of the original colour image, followed by a multi-scale reintegration to obtain the final daltonised image. Although giving good results, the method had two main problems: (i) it was very computationally expensive, and (ii) it tended to produce visual halo artefacts near strong chromatic edges. In the present paper, we present a solution to these two problems by (i) finding a much better initial condition for the iterative algorithm, and (ii) introducing local linear anisotropic diffusion in place of the non-local isotropic diffusion used in the previous method. Together this leads to significantly shorter computing times, as well as to the elimination of visual halo artefacts while still preserving the qualitative behaviour of the previously proposed algorithm.

## 2. Background

In the previously proposed algorithm [17], the original image was represented as the vector-valued function
$$u_{0}:\Omega \to C,$$
where $\Omega \subset R^{2}$ was the spatial image domain, and $C\subset R^{3}$ the colour space. The spatial gradient $\nabla u_{0}$ represented the two spatial directions for all image channels [19]. The CVD simulated image using the method of Brettel et al. [4], was denoted $s(u_{0})$. The difference image between the original image and its simulation, $d_{0}=u_{0}-s\left(u_{0}\right)$, contained the information that is lost for CVD observers.

We applied a principal component analysis (PCA) [20] of $d_{0}$ to obtain its first principal component $p_{1}\left(d_{0}\right)$, representing the main diretion of difference, i.e., the main direction of information loss for CVD observers. An orthonormal basis for the daltonisation was built using the Gram–Schmidt process starting from the normalised grey axis taken as either the diagonal of the RGB space, $e_{l}=\left[1,1,1\right]$, or as the direction orthogonal to the plane of constant lightness in, e.g., sRGB space, $e_{l}=\left[0.2126,0.7152,0.0722\right]$ [21], and $p_{1}\left(d_{0}\right)$. This resulted in the orthonormal triplet $e_{l}$, $e_{d}$, and $e_{c}$, representing the direction of lightness, difference, and maximally visible chroma change of the CVD observer, respectively.

The gradient of the difference between the original image and the result of the simulation algorithm was then approximated by the projection of the image gradient onto $e_{d}$, $\nabla d_{0}\approx \left(\nabla u_{0}\cdot e_{d}\right)e_{d}$ (for a simulation algorithm that is an orthogonal projection in an equi-luminant plane of the colour space, the approximation is exact). This difference was rotated and scaled into the direction of maximal chromatic visibility, $e_{c}$, and added to the original gradient field to obtain a new gradient tensor
(1)$$G=\nabla u_{0}+\chi \left(\nabla u_{0}\cdot e_{d}\right)e_{c},$$
where χ was a suitable scalar field.

In order to determine χ, we assumed the norm of the simulated tensor S(G) to be equal to the norm of the original gradient because the visible gradient for a colour-deficient observer looking at the daltonised image should equal in magnitude the gradient for a normal-sighted observer looking at the original image,
(2)$${∥S(G)∥}_{F}^{2}={∥\nabla u_{0}∥}_{F}^{2}.$$

Here, ${∥\ldots ∥}_{F}^{2}$ represents the Frobenius norm over both the colour space and the spatial gradient space simultaneously. For simplicity, we approximated the simulation method by a linear projection onto the $\left(e_{c},e_{l}\right)$ plane, $S\left(G\right)\approx \nabla s\left(u_{0}\right)+\chi \left(\nabla u_{0}\cdot e_{d}\right)e_{c}$. Inserting this into Equation (Equation 2) gives
$$\begin{array}{cc} \chi^{2}{∥\left(\nabla u_{0}\cdot e_{d}\right)e_{c}∥}_{F}^{2} & \\ +2\chi \left(\nabla u_{0}\cdot e_{d}\right)e_{c}:\nabla s\left(u_{0}\right) & \\ +{∥\nabla s(u_{0})∥}_{F}^{2}-{∥\nabla u_{0}∥}_{F}^{2} & =0, \end{array}$$
a quadratic equation for χ with, in general, two solutions, $\chi_{\pm }$. Which one to use was found by integrating both solutions over the image, selecting the smaller one in absolute value.

Finally, the daltonised image from the tensor G was found by reintegrating the gradient field iteratively through gradient descent [22] using the finite difference method with explicit time integration—hence referred to as the *isotropic* diffusion equation
(3)$$\frac{\partial u}{\partial t}=\nabla \cdot \left(\nabla u-G\right),$$
with a suitable stopping criterion. Further fine tuning was made using a fidelity term to preserve memory colours such as skin tones.

The original image $u_{0}$ was used as an initial condition for the gradient descent. Since the daltonisation process in some cases lead to a significant change of the colours of the image, and since high image resolution gave very long diffusion paths, the algorithm took very long to converge. In order to reduce the computing time, a multi-scale pyramidal technique was used, first solving the daltonisation problem for a very small down-scaled image, then iteratively using the result as a starting point for the solution at a higher resolution until the solution for the full scale image was found.

## 3. Proposed Method

### 3.1. The Initial Value

One main problem of the previously proposed algorithm is the computation time, mainly stemming from long diffusion paths and the use of the original image $u_{0}$ as the initial value. Like for all optimisation problems, the selection of a good initial guess can be crucial for the time of convergence.

Instead of using the original colour image $u_{0}$ as the initial value, we propose here to perform an operation similar to the one applied above to the gradient directly in the colour domain. Letting $s(u_{0})$ denote the simulated image, and constructing $e_{l}$, $e_{d}$, and $e_{c}$ as before, the colour information lost by the CVD observer represented by the simulation algoritm can be approximated by the projection of the original image onto $e_{d}$, i.e., $d_{0}\approx \left(u_{0}\cdot e_{d}\right)e_{d}$. Rotating this estimate of the lost information into the direction of maximal chromatic visibility and clipping to the colour gamut gives us a much better first estimate of the daltonised image,
(4)$$u_{s}=g\left(u_{0}+\left(u_{0}\cdot e_{d}\right)e_{c}\right),$$
where g(·) is the gamut clipping operator. Actually, this simple operation works well as a simple daltonisation method all by itself. It produces a recolouring similar to the one by the previously proposed algorithm, but destroys quite some image features and textures due to the gamut clipping operation. However, no halo artefacts are introduced. See Section 4 for sample results.

### 3.2. Local Linear Anisotropic Diffusion

The second significant problem of the previously proposed method is the production of visible halo artefacts near strong chromatic edges in the images. Switching from non-local isotropic to local linear anisotropic diffusion has proven to be a strong method to reduce haloing problems in various PDE based colour imaging solutions such as colour gamut mapping [23], colour-to-greyscale image conversion [24], and colour image demosaicing [25]. Thus, we propose to apply the same technique here.

The components of the structure tensor S of the original image can be expressed as [26]
$$S_{ij}=\sum_{\mu ,\nu }\frac{\partial u_{0}^{\mu }}{\partial x^{i}}\frac{\partial u_{0}^{\nu }}{\partial x^{j}}\;.$$

The eigenvalues of S are denoted $\lambda^{+}$ and $\lambda^{-}$, and the corresponding normalised eigenvectors $e^{+}$ and $e^{-}$ are stored as columns in the orthonormal eigenvector matrix E, such that the structure tensor can be written $S=E^{T}\mathrm{diag}\left(\lambda^{+},\lambda^{-}\right)E$.

From this, the diffusion tensor can, in agreement with Sapiro and Ringach [27], be defined as
$$D=E^{T}\mathrm{diag}\left(d\left(\lambda^{+}\right),d\left(\lambda^{-}\right)\right)E,$$
where d(λ) is a nonlinear diffusion coefficient function,
$$d\left(\lambda \right)=\frac{1}{1+\kappa \lambda^{2}}\;,$$
and κ is a suitably chosen numeric constant.

Inserting this into Equation (Equation 3), gives the *anisotropic* diffusion equation
(5)$$\frac{\partial u}{\partial t}=\nabla \cdot \left[D\left(\nabla u-G\right)\right]$$
subject to the constraint u∈C with the initial condition $u\left(t=0\right)=u_{s}$.

### 3.3. Further Simplifications

One complicating factor in the previously proposed method is the computation of the scalar field χ and the selection of the best solution for it. Through experimentation, we have found that we obtain at least visually equally good daltonised images by simply setting χ=1 thorughout, i.e., simply removing χ from Equation (Equation 1) giving
$$G=\nabla u_{0}+\left(\nabla u_{0}\cdot e_{d}\right)e_{c}.$$

A second complicating factor in the previously proposed method was the introduction of the data attachment term for certain memory colours. Since the main purpose of daltonisation methods as such is behavioural, i.e., making observers able to perform certain tasks with the images, we propose here to skip that term altogether.

### 3.4. Implementation

The methods are discretised using the finite difference method with explicit time integration, and implemented in Python using the NumPy, Scipy, and Matplotlib libraries. The code is freely available on GitHub (https://github.com/ifarup/anisotropic-daltonisation/).

## 4. Results

In order to test the proposed algorithm, we gathered a small set of colourful images from the web with dominating red–green and blue–yellow contrasts. All the gathered images are freely available under the CC0 license (https://creativecommons.org/share-your-work/public-domain/cc0/). The images are shown in Figure 1.

Since the algorithm proposed above works for any kind of CVD simulation algorithm, we could have used the dichromat simulation method by Brettel et al. [4] like in [17]. However, in order to test the behaviour for various kinds of dichromats and anomalous trichromats, we develop a simple parametrised version of their method. The simulation is performed directly in the RGB colour space using the two dichromatic matrices,
$$M_{\mathrm{RG}}=\left(\begin{array}{ccc} 1/2 & 1/2 & 0\\ 1/2 & 1/2 & 0\\ 0 & 0 & 1 \end{array}\right)\;\mathrm{and}\;M_{\mathrm{BY}}=\left(\begin{array}{ccc} 1/2 & 0 & 1/2\\ 0 & 1/2 & 1/2\\ 1/4 & 1/4 & 1/2 \end{array}\right),$$
for red–green (RG) and blue–yellow (BY) dichromats, respectively. In order to simulate also anomalous trichromats, we weigh in the orinal image with a certain weight α
s(u)=αMu+(1−α)u,
where *M* is either $M_{\mathrm{RG}}$ or $M_{\mathrm{BY}}$, such that α=0 corresponds to colour normal observes, 0<α<1 to anomalous trichromats, and α=1 to dichromats.

We run the images through the three following daltonisation algorithms:the simple daltonisation algorithm proposed in Equation (Equation 4),the isotropic daltonisation algorithm proposed in Equation (Equation 3), which is essentially the same as the one proposed in [17], but with the simplifications described in Section 3.3, and the simple daltonisation as the initial condition, andthe anisotropic daltonisation algorithm proposed in Equation (Equation 5).

This is performed for four different simulated CVD observers: RG and BY with α=1.0 and α=0.8, resulting in a total of 14×3×4=504 daltonised images. Typical results for one original image and one observer (RG dichromat with α=1) are shown in Figure 2.

Zooming into a detail of the sample image, we see in Figure 3 that the simple daltonisation algorithm leads to a significant loss of details. This is caused by the the clipping of colours that end up being mapped outside the RGB gamut. The anisotropic daltonisation, on the other hand, is able to preserve much more of the image details. The difference can be quantified by looking at the gamut pixel fraction (*GPF*), i.e., the fraction of the image pixels that reside on the gamut boundary, indicated by having pixel values of 0 or 1,
$$\mathrm{GPF}=\frac{\#(\mathrm{pixels}\mathrm{on}\mathrm{the}\mathrm{gamut}\mathrm{boundary})}{\#(\mathrm{pixels}\mathrm{in}\mathrm{the}\mathrm{image})}.$$

The resulting *GPF* values for all the images are shown as a box plot in Figure 4. Although the distributions for isotropic and anisotropic daltonisation are overlapping, they are statistically significantly different according to the Wilcoxon signed-rank test with a *p*-value $p=6.7\times 10^{-8}$, and with a median in favour of the anisotropic daltonisation.

As noted already in the previous paper [17], visible halo artefacts are observed for the isotropic daltonisation method in many of the images. One such example is shown in Figure 5. The anisotropic daltonisation, on the other hand, produces far less halo artefacts. This can be quantified by comparing how similar the various daltonised images are to the original images by measuring the *PSNR* values,
$$\begin{array}{cc} \mathrm{PSNR} & =-10\log(\mathrm{MSE}),\\ \mathrm{MSE} & =\frac{1}{MN}\sum_{i=0}^{M-1}\sum_{j=0}^{N-1}{\left[I\left(i,j\right)-K\left(i,j\right)\right]}^{2}, \end{array}$$
where *I* and *K* indicate the two images that are compared, and *M* and *N* denote the horisontal and vertical image resolutions, respectively. The resulting *PSNR* values for all the images are shown as a box plot in Figure 6. Although the distributions for isotropic and anisotropic daltonisation are overlapping, they are statistically significantly different according to the Wilcoxon signed-rank test with a *p*-value of $p=7.5\times 10^{-11}$, and with a median in favour of the anisotropic daltonisation.

## 5. Conclusions

We have developed a new daltonisation algorithm based on the previously developed method presented in [17]. Unlike the previous method, the proposed method is individualised in the sense that it can be used successfully with any CVD simulation method for dichromats and anomalous trichromats. The method is computationally much more efficient than the previously proposed method due to the simple daltonisation algorithm that acts as a much improved initial value for the iterative numerical scheme. The resulting images are free of the halo artefacts seen previously due to the introduction of a local linear anisotropic diffusion replacing the non-local isotropic diffusion of the previous method. As such, it is the first spatially dependent artefact-free daltonisation algorithm known to the authors. Comparison of the proposed algorithm with other recent state-of-the-art daltonisation methods in behavioural and psychometric experiments is left for future research.

## Figures and Tables

**Figure 1 jimaging-06-00116-f001:**
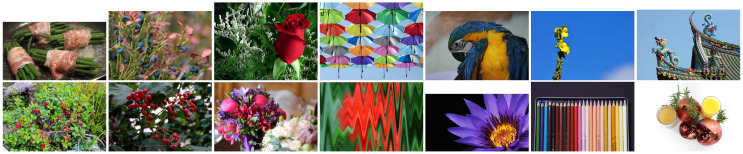
The image dataset used for testing. Highly saturated images from the web with dominant red–green and blue–yellow contrast. All images are available under the CC0 licence.

**Figure 2 jimaging-06-00116-f002:**
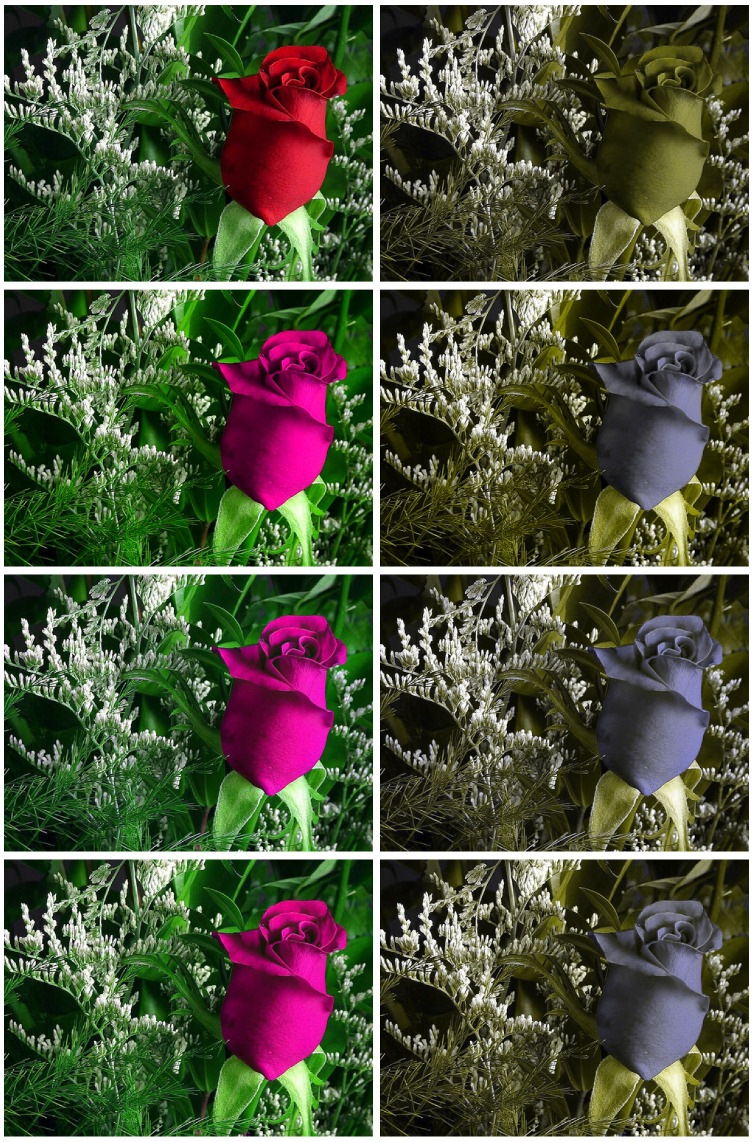
Sample image for an RG α=1 dichromatic observer. (**Top** to **bottom**, **left** hand column): original, simple daltonisation, isotropic daltonisation, and anisotropic daltonisation. Corresponding simulated images for a the RG α=1 dichromatic observer in the right hand column.

**Figure 3 jimaging-06-00116-f003:**
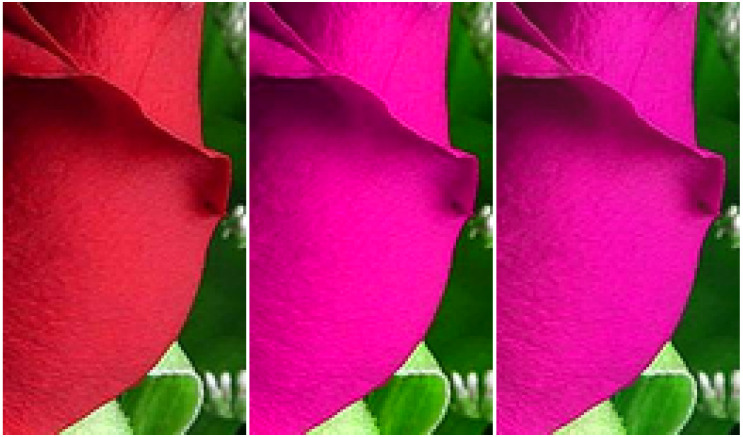
Loss of details in the simple daltonisation algorithm for an RG α=1 dichromatic observer. **Left** to **right**: original image, simple daltonisation, and anisotropic daltonisation.

**Figure 4 jimaging-06-00116-f004:**
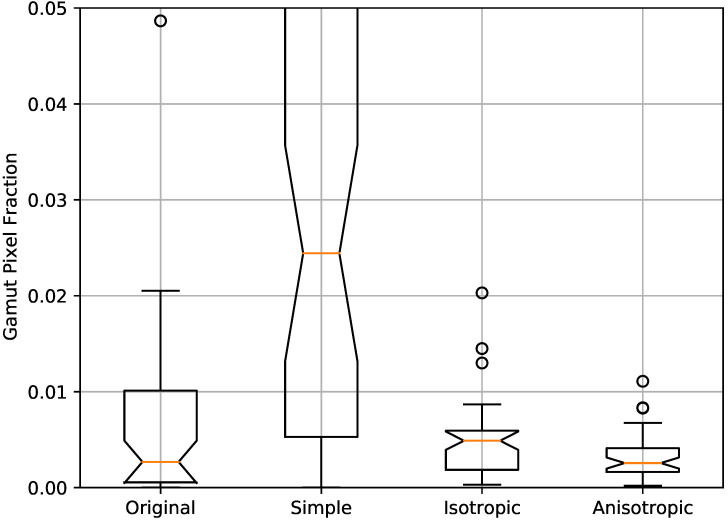
Box plot of the gamut pixel fraction (*GPF*) for the original and the simple, isotropic, and anisotropic daltonised images.

**Figure 5 jimaging-06-00116-f005:**
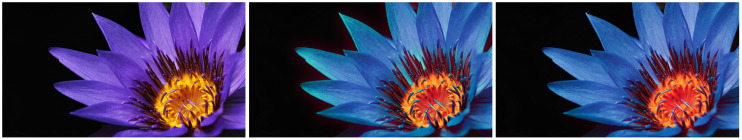
Halo artefacts in the isotropic daltonisation algorithm for a BY α=1 observer. (**Left** to **right**): original image, isotropic daltonisation, and anisotropic daltonisation.

**Figure 6 jimaging-06-00116-f006:**
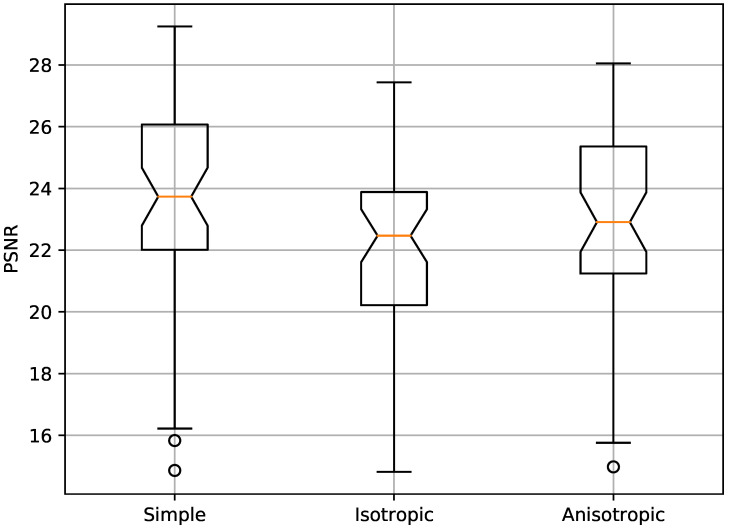
*PSNR* (dB) of the daltonised versus the original images for the simple, isotropic, and anisotropic daltonisation algorithms.

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
