# Peer review of "Individualised Halo-Free Gradient-Domain Colour Image Daltonisation"

_2313-433X, 2020, doi:10.3390/jimaging6110116_

Round 1

Reviewer 1 Report

A short paper with a small incremental contribution, but the claim is not over-stated and the small improvement seems reasonably clear. The paper is well written and well reasoned. Results are I think appropriate for the scope of the work and contributions. It may have been nice to see some figures for the performance improvement achieved but from a theoretical standpoint, I'm convinced. 

I think it would be ok to accept.

Author Response

Thank you for a nice review. I have not made any changes to the manuscript based on these comments.

Reviewer 2 Report

This manuscript proposes an improved color image daltonisation method. By implementing an estimation of the daltonised image and local linear anisotropic diffusion, the halo artefacts could be removed, and the efficiency is largely improved.

Major comments:

  1. In the introduction chapter, the most recent related works (after year 2018) on this topic are missing.

Minor comments:

  1. A few words (and equations) should be added to describe the gamut pixel fraction (GPF) and PSNR
  2. Some typos, e.g., Line 66: Which on to use was found by -> one

Author Response

Thank you for the thorough review. Major comments:

  1. Thanks for pointing out this. I have redone my literature review and added a few new relevant references to the introduction.

Minor comments:

  1. Comments and equations for GPF and PSNR have been added
  2. The typo was already corrected by the publisher

Reviewer 3 Report

The manuscript entitled “Individualised Halo-Free Gradient-Domain Colour Image Daltonisation” seems to fit the aims of the scientific journal J. Imaging. The topic of the study is interesting and it fits into trends in science as well as in the manufacturing practice. Given the scope of the results presented, it is necessary to improve conclusion section. Please rewrite it clearly stating the facts; focus more on how your research has contributed to knowledge gaps; describe research limitations for future research and restate your major findings; add scientific and practical significance of the selected method.

Author Response

Thank you for the thorough review. The conclusion has been expanded in agreement with your proposal.